# Loss of Function *TGFBR2* Variant as a Contributing Factor in Generalized Pustular Psoriasis and Adult-Onset Immunodeficiency

**DOI:** 10.3390/genes14010103

**Published:** 2022-12-29

**Authors:** Piranit Kantaputra, Teerada Daroontum, Mati Chuamanochan, Suteeraporn Chaowattanapanit, Worrachet Intachai, Bjorn Olsen, Thanapat Sastraruji, Sissades Tongsima, Chumpol Ngamphiw, Jatupol Kampuansai, Timothy C. Cox, Salin Kiratikanon

**Affiliations:** 1Center of Excellence in Medical Genetics Research, Faculty of Dentistry, Chiang Mai University, Chiang Mai 50200, Thailand; 2Division of Pediatric Dentistry, Department of Orthodontics and Pediatric Dentistry, Faculty of Dentistry, Chiang Mai University, Chiang Mai 50200, Thailand; 3Department of Pathology, Faculty of Medicine, Chiang Mai University, Chiang Mai 50200, Thailand; 4Division of Dermatology, Department of Internal Medicine, Faculty of Medicine, Chiang Mai University, Chiang Mai 50200, Thailand; 5Division of Dermatology, Department of Medicine, Faculty of Medicine, Khon Kaen University, Khon Kaen 40000, Thailand; 6Department of Developmental Biology, Harvard School of Dental Medicine, Boston, MA 02115, USA; 7Dental Research Center, Faculty of Dentistry, Chiang Mai University, Chiang Mai 50200, Thailand; 8National Biobank of Thailand, National Science and Technology Development Agency (NSTDA), Thailand Science Park, Pathum Thani 12120, Thailand; 9Department of Biology, Faculty of Science, Chiang Mai University, Chiang Mai 50200, Thailand; 10Departments of Oral & Craniofacial Sciences and Pediatrics, School of Dentistry and School of Medicine, University of Missouri-Kansas City, Kansas City, MO 64108, USA

**Keywords:** adult-onset immunodeficiency syndrome, anti-interferon-γ autoantibody, TGFBR2 mutation, generalized pustular psoriasis, predisposing risk factor, pustular skin reaction

## Abstract

Background: Generalized pustular psoriasis (GPP; MIM 614204) is a rare multisystemic autoinflammatory disease, characterized by episodes of acute generalized erythema and scaling developed with the spread of numerous sterile pustules. Adult-onset immunodeficiency syndrome (AOID) with anti-interferon-γ autoantibodies is an immunodeficiency disorder associated with disruptive IFN-γ signaling. Methods: Clinical examination and whole exome sequencing (WES) were performed on 32 patients with pustular psoriasis phenotypes and 21 patients with AOID with pustular skin reaction. Histopathological and immunohistochemical studies were performed. Results: WES identified four Thai patients presenting with similar pustular phenotypes—two with a diagnosis of GPP and the other two with AOID—who were found to carry the same rare TGFBR2 frameshift mutation c.458del; p.Lys153SerfsTer35, which is predicted to result in a marked loss of functional TGFBR2 protein. The immunohistochemical studied showed overexpression of IL1B, IL6, IL17, IL23, IFNG, and KRT17, a hallmark of psoriatic skin lesions. Abnormal TGFB1 expression was observed in the pustular skin lesion of an AOID patient, suggesting disruption to TGFβ signaling is associated with the hyperproliferation of the psoriatic epidermis. Conclusions: This study implicates disruptive TGFBR2-mediated signaling, via a shared truncating variant, c.458del; p.Lys153SerfsTer35, as a “predisposing risk factor” for GPP and AOID.

## 1. Introduction

Generalized pustular psoriasis (GPP; MIM 614204), the most severe form of all the psoriatic diseases, is a rare multisystemic autoinflammatory disease characterized by episodes of acute generalized erythema and scaling developed with the spread of numerous sterile pustules [1]. Patients with GPP may have pustular skin reactions similar to those seen in individuals with adult-onset immunodeficiency syndrome (AOID) with anti-interferon-γ (IFN-γ) autoantibodies [1,2].

AOID is an autoimmune disease characterized by an AIDS-like illness with abnormal IFN-γ/IL12 signaling. The clinical manifestations of patients affected with AOID resemble those in patients with inborn deficiencies of IFN-γ production or IFN-γ-response because IFN-γ has a non-redundant role in fighting mycobacterial infection [2,3]. Patients with AOID generally present with signs and symptoms of aberrant cell-mediated immunity including recurrent and disseminated infection by mycobacteria with low virulence. Their clinical responses to anti-mycobacterial therapy are generally poor [4]. The majority of cases have skin involvement, including reactive skin disorders (82%) and infective skin diseases (45%), which are mostly neutrophilic dermatoses [2,5]. AOID is more prevalent in Thailand and Taiwan than the rest of the world [2], suggestive of a genetic predisposition in these populations. The age of onset of AOID is around 30–50 years old [2]. AOID shares skin manifestations with GPP, including the pustular reaction, suggestive of shared pathogenetic mechanisms [5].

To the best of our knowledge, however, AOID and GPP have not been reported in the same subjects or within the same families. Regarding genetic susceptibility, HLA-DRB1*15:01, DRB1*16:02 and HLA-DQB1*05:01, DQB1*05:02 alleles have been reported to be associated with AOID in the Thai population [4], whereas variants in other genes including *IL36RN, CARD14, IL1RN, AP1S3, MPO, TNIP1*, and *SERPINA3* have been reported to be associated with GPP [1], [Genovese et al., 2021]. Recently, we reported heterozygous variants in *SERPINA3* and *SERPINA1* in patients with AOID [6] and GPP [7], respectively.

Transforming growth factor beta (TGFB) signaling has important roles in regulating a variety of cellular processes as well as the formation of extracellular matrix. TGFB1 has an anti-proliferative role in the epidermis [8]. Mechanistically, TGFB1 ligand binds first to TGFBR2 (TRANSFORMING GROWTH FACTOR-BETA RECEPTOR, TYPE II; MIM 190182), followed by the interaction and phosphorylation of a glycine-serine-rich domain of TGFBR1 (TRANSFORMING GROWTH FACTOR-BETA RECEPTOR, TYPE I; MIM 190181) to form an activated ligand–receptor complex. Both TGFBR1 and TGFBR2 are transmembrane serine/threonine kinases. Transphosphorylation of the glycine–serine-rich domain of TGFBR1 by TGFBR2 switches the glycine–serine-rich domain from the binding site for the inhibitor FKBP12 to a binding site for the R-Smad substrate, initiating Smad-dependent signal transduction [9]. TGFBR2 is constitutively active [9], and its expression is dramatically decreased or absent in psoriatic skin [8]. Variants in *TGFBR2* have been reported to be associated with Loeys-Dietz syndrome 2 (MIM 610168), colorectal cancer-hereditary nonpolyposis type 6 (MIM 614331), and esophageal cancer (MIM 133239) [10,11]. Variants in *TGFBR2* have not previously been reported to be associated with GPP and AOID. Previously, a genetic variant in SERPINA1 was found to be a predisposing factor in the presentation of both GPP and AOID [7]. Although TGFBR2 has not previously been reported to be associated with GPP and AOID, the expression of TGFBR2 is notably decreased or absent in psoriatic skin [8]. Therefore, we hypothesized that genetic variants in TGFBR2 might also predispose some patients in our cohort to GPP and AOID.

Here, we report four Thai patients, two presenting with GPP and two AOID with a pustular skin reaction, who carried the same heterozygous frameshift mutation in TGFBR2.

## 2. Patients and Methods

### 2.1. Patients

This study involving human participants was approved by the Human Experimentation Committees of the Faculty of Dentistry, Chiang Mai University (no. 71/2020), the Faculty of Medicine, Chiang Mai University and the Faculty of Medicine, Khon Khan University and was performed in accordance with the ethical standards of the 1964 Declaration of Helsinki and its later amendments or comparable ethical standards. Informed consent was obtained from all participants.

Clinical examination and whole exome sequencing were performed on our cohort of 53 patients, including 32 patients with pustular psoriasis phenotypes and 21 patients with AOID with pustular skin reaction. Inclusion criteria were patients with AOID or pustular psoriasis phenotypes, including GPP, acrodermatitis continua of Hallapeau (ACH), and palmoplantar pustulosis (PPP). Exclusion criteria were patients without AOID or pustular psoriasis phenotypes. Whole exome sequencing showed a heterozygous frameshift variant in TGFBR2 (c.458del; p.Lys153SerfsTer35) in two patients with GPP and two patients with AOID and they are present here.

#### 2.1.1. Patient 1

A 70-year-old Karen tribe woman had presented with non-follicular pustules on erythrodermic skin of her trunk and extremities since the age of 50. Nail psoriasis was also observed (Figure 1A–D,F). She had episodes approximately once a year. Diagnosis of GPP with erythroderma was made. No specific environmental or dietary triggers were noted. The rashes were well-controlled with rotational methotrexate and oral acitretin treatment. A skin biopsy from pustules on her abdomen showed subcorneal and spongiform pustules (Figure 1E).

#### 2.1.2. Patient 2

A 27-year-old Thai woman had developed recurrent episodes of generalized pustular eruption for 3 years. She was then referred to Maharaj Nakorn Chiang Mai Hospital for definite diagnosis. She was afebrile, and dermatological examination showed multiple non-follicular pustules on the erythrodermic skin of her trunk and extremities. Microbiological studies from pustules failed to demonstrate any organisms. Diagnosis of relapsing GPP without systemic inflammation was made according to diagnostic criteria (Navarini et al., 2017) [12]. The pustules were well-controlled with oral acitretin.

#### 2.1.3. Patient 3

A 66-year-old man presented with generalized lymphadenitis as a result of possible disseminated non-tuberculosis mycobacterial (NTM) infection (evidenced from lymph node biopsy revealing granulomatous lymphadenitis and responded well to NTM treatment regimen). During that time, he also developed sterile generalized non-follicular pustules on his trunk and extremities which responded well to oral acitretin. Diagnosis of AOID with pustular reaction was made (Figure 2).

#### 2.1.4. Patient 4

A 41-year-old previously healthy Thai female presented with a 6-month history of fever with generalized lymphadenopathy and weight loss. One week before admission at Srinagarind hospital, she developed pustular eruption on both hands (Figure 3). Cervical lymph node biopsy revealed sclerotic lymphoid hyperplasia with plasmacytosis and a focus of eosinophilic abscess. Although no organisms were detected on histopathology, *Mycobacterium abscessus* was detected using PCR and isolated from cultures of the tissue. Gram and Ziehl Neelsen stains from pustules on the hands revealed no organisms, and cultures were negative. The result of an HIV test was negative, but a test for anti-IFN-γ autoantibodies was positive at 1:5000. A diagnosis of AOID with pustular eruption was made. Anti-mycobacterial therapy with azithromycin, levofloxacin, and prednisolone was initiated. Her symptoms, generalized lymphadenopathy, and pustular eruption improved after two weeks of treatment. Anti-mycobacterial therapy was continued for 20 months with remarkable clinical improvement.

### 2.2. Whole Exome Sequencing and Mutation Analysis

Genomic DNA was extracted, with consent, from whole blood following standard procedures. The DNA samples of the patients were subject to whole exome sequencing following processing using a SureSelect V6+UTR-post target capture kit. Genomics analysis toolkit (GATK) germline mutation workflow version 3.8.1 (Cambridge, IL, USA) [13] was utilized to identify variants. The sequencing reads were aligned to the human genome reference (hg19) using BWA-MEM version 0.7.17 [13,14] to generate BAM files. These BAM files were processed by GATK HaplotypeCaller to identify SNVs and small indels resulting in individual GVCF files. These GVCF files were consolidated into a single joint genotyped VCF file format, listing all genotypes in separate columns. Ensembl variant effect predictor tool (version 95) [15] was used to predict pathogenic effects of each variant.

### 2.3. Histopathology

Biopsies were taken from the lesion and immediately fixed in a 10% neutral-buffered formalin solution. After fixation, dehydration, clearing, and infiltration by paraffin wax, the samples were embedded in the paraffin block. For histopathological examination, paraffin-embedded tissues were sectioned at a thickness of 3 µm and stained with hematoxylin and eosin (H&E) using standard histological laboratory methods. Histopathologic characterization of the specimens was performed by experienced dermatologists and dermatopathologists.

### 2.4. Immunohistochemistry

Formalin-fixed paraffin-embedded tissues were sectioned at a thickness of 3 μm and mounted on Superfrost plus microscope slides. Slides were heated for 1 h at 60 °C in a dry oven to facilitate attachment of tissue and soften the paraffin. Immunohistochemical staining was performed on a Ventana BenchMark ULTRA autostainer using a standard established protocol.

In brief, the sections were deparaffinized, rehydrated, and antigen retrieved using CC1 (prediluted, PH 8.0) antigen retrieval solution (Ventana) performed on the Benchmark ULTRA automated slide Stainer. The sections were incubated with primary antibodies, at the manufacturer’s recommended dilution. The following primary antibodies were used: rabbit monoclonal anti-human TGFB1 [clone EPR21143] (ab215715, Abcam, Cambridge, MA, USA, 1:500 dilution), rabbit monoclonal anti-human KRT17 [clone EPR1624Y] (ab51056, Abcam, Cambridge, MA, USA, 1:400 dilution), mouse monoclonal anti-human IL1B [2H12] (sc130323, Santa Cruz Biotechnology, Inc., Dallas, TX, USA, 1:50 dilution), rabbit polyclonal anti-human IL17A (ab79056, Abcam, Cambridge, MA, USA, 1:100 dilution), rabbit monoclonal anti-human IFN-γ [IFNG] (ab218426, Abcam, Cambridge, MA, USA, 1:50 dilution), rabbit polyclonal anti-human IL6 (ab6672, Abcam, Cambridge, MA, USA, 1:100 dilution), mouse monoclonal antibody anti-human IL-23 [clone HLT2736] (BioLegend, San Diego, CA, USA, 1:100 dilution), and mouse monoclonal anti-human TNFA [Clone 28401] (mab610, R&D Systems, Minneapolis, MN, USA, 1:100 dilution). The Ultraview universal DAB IHC detection kit was used for the visualization reaction, then counterstained with hematoxylin and bluing reagent. The slides were then gently washed, dehydrated in graded ethanol and xylene, and a coverslip mounted using mounting media (Sigma-Aldrich, Taufkirchen, Germany).

## 3. Results

### 3.1. Whole Exome Sequence Sequencing and Bioinformatic Analysis

Whole exome sequence analysis of each case did not reveal any candidate genes with autosomal recessive inheritance. However, a heterozygous frameshift variant in TGFBR2 (chr3: g.30691880del; c.458del; p.Lys153SerfsTer35) was identified and confirmed by Sanger direct sequencing in all four unrelated patients (Figure 4). This variant is not seen in over 250,000 alleles reported in gnomAD v2.1.1 (https://gnomad.broadinstitute.org/ (accessed on 14 November 2022)), although a frameshift impacting the same codon, c.459del; p.Lys153AsnfsTer35, has been seen in one person of South Asian ancestry. In total, only 11 frameshift and five possible splice site variants have been reported in TGFBR2 in gnomAD.

The c.458del; p.Lys153SerfsTer35 variant was found in 17 individuals in our in-house exome database of 725 individuals of Thai ancestry with normal controls who had no history of pustular skin lesions. Thirty-six additional Thai patients with either GPP or AOID did not carry the truncating TGFBR2 variant as determined by whole exome sequencing. Fisher’s exact test revealed a statistically significant difference in the variant frequency between the diseased cases and control groups: 10.0% vs. 2.3%, respectively (*p* = 0.020). In addition to the truncating TGFBR2 variant, an additional variant in TGFBR2 (NM_001024847.2: c.1019C>T; NP_001020018.1: p.Thr340Met; rs34833812) was also identified in patient 2. The allele frequency for this variant is 0.001062, according to gnomAD, and was detected in four of 725 normal Thai controls who had no history of pustular skin lesions. This variant has been classified by ClinVar as benign based on supporting modeling and functional data.

In addition to the TGFBR2 variants, rare variants in other immunodeficiency genes were also found in the four cases reported here (Table 1). Of note, the CARD14 variant c.2473G>A; p.Ala825Thr found in patient 1, who was from the Karen tribe, was also seen in 90 of 100 Karen tribe individuals we investigated by Sanger sequencing, suggestive of a polymorphism.

### 3.2. Histopathological Findings

Histological analysis of skin biopsies from patients 1 and 3 revealed notable pathology. In patient 1 with GPP, lesional skin histology revealed sub-corneal and intra-epidermal collections of neutrophils. Neutrophil microabscesses were observed in the stratum spinulosum and stratum corneum on a background of marked acanthosis and elongated rete ridges. Similar neutrophil-rich macroscopic pustules were noted underneath the stratum corneum in patient 3 (AOID), along with epidermal acanthosis, elongation of rete ridges, and inflammatory infiltrate (Figure 5).

### 3.3. Immunohistochemical Findings

Immunohistochemical investigation showed overexpression of IL6, IL17, KRT17, and IFNG in psoriatic skin lesions of patients 1 and 3. TGFB1 staining was elevated within the collections of neutrophils in both the epidermis and some dermal cells in patient 3 (Figure 6). IL1B and IL23 showed mildly increased expression in the dermis of patient 3. There was no significant difference of TNFA expression between normal skin tissue and patients 1 and 3 (Figure 7).

## 4. Discussion

Germline pathogenic variants in *TGFBR2* have been reported to underlie Loeys-Dietz syndrome, an autosomal dominant Marfan syndrome-like connective tissue disorder characterized by hypertelorism, cleft palate with bifid uvula, joint laxity, scoliosis and arachnodactyly, translucent and easily bruised skin, and severe aortic aneurysms [10]. In the vast majority of the above cases, the pathogenic variants are missense mutations, with most residing in the cytoplasmic kinase domain [10]. Somatic mutations in *TGFBR2* are also commonly seen in hereditary non-polyposis colorectal cancer (HNPCC) with defects in mismatch repair (MMR). Tissues harboring defects in MMR exhibit microsatellite instability (MSI) that is characterized by slippage of short to mid-length mononucleotide repeats (or homopolymers) that cannot efficiently repair in the absence of MMR [11]. Because of a polyA tract in exon 3, *TGFBR2* is one of the most frequent targets of MSI: ~90% of HNPCC cases harbor frameshift mutations in TGFBR2 [10]. Notably, this is the same polyA tract affected by the frameshift variant, c.458del; p.Lys153SerfsTer35, in each of our patients (two with GPP and the other two with AOID). Because of the location of this frameshift variant within the *TFGBR2* locus, it is expected that transcripts derived from the variant *TGFBR2* allele in our patients would undergo nonsense-mediated decay, resulting in decreased TGFBR2 protein and reduced TGFβ signaling, and thus potentially a different mechanistic impact to those variants seen in Loeys-Dietz syndrome. Even if some truncated protein was produced and reached the plasma membrane, it would likely be secreted as the truncation occurs before the transmembrane domain. Such truncated and secreted type II receptors can act in a dominant-negative manner by preventing the interaction of the remaining functional type II receptor with type I receptor [16]. A third possibility is that the mutant protein does not make it out of the ER, thus reducing the overall surface levels of active TGFBR complexes.

### 4.1. TGFB Signaling, Its Antiproliferative Effect, and GPP

TGFβ signaling is known for its anti-proliferative and immunosuppressive effects [8]. Thus, the hyperproliferation of psoriatic epidermis seen in patients 1 and 3 is consistent with disrupted TGFβ signaling as a result of a heterozygous loss of function *TGFBR2* mutation. In psoriatic epidermis, aberrant TGFβ signaling has been shown to lead to downregulation of the microRNA, miR-486-3p, which in turn leads to overexpression of KRT17, a hallmark of psoriatic lesions. Concomitant with the overexpression of KRT17 is hyperproliferation of the keratinocytes and subsequent neutrophil infiltration [17] (Figure 6). Overexpression of KRT17 found in patients 1 and 3 is therefore also consistent with a role for reduced TGFBR2 in GPP and AOID in patients 1 and 3, respectively (Figure 6). This neutrophil infiltration results in overproduction of pro-inflammatory cytokines, including IL17A, IL22, IFN-γ, IL6, and TNF𝜶 from Th1 and Th17 cells, which can result in subsequent tissue destruction. This is supported by the increased expression of IL1B, IL6, IL17, IL23, and IFNG in the lesional skin of patients 1 and 3 (Figure 6). Aberrant TGFβ signaling might stimulate keratinocytes to produce IL23 and induce inflammatory dendritic cells into evolving psoriatic lesions [18]. Interestingly, we did not find changes in staining for the inflammatory cytokine, TNFA (Figure 7).

### 4.2. TGFB Signaling, Immunosuppressive Effects, and AOID

IFN-γ is an important cytokine produced by activated T-cells (Th1), natural killer cells, and group 1 innate lymphoid cells [3]. IFN-γ has a crucial and non-redundant role in defending against mycobacterium infection [3]. The IFN-γ/IL-12 pathway signaling, which predominantly activates monocytes, is important to prevent invasion of mycobacteria and other intracellular pathogens and reactivation of latent varicella-zoster virus [5]. This pathway is disrupted in patients affected with AIDS and AOID [19], which makes individuals susceptible to mycobacterial and viral infection.

TGFβ signaling normally promotes the differentiation of naïve T cells into Th17 cells [20]. Aberrant TGFβ signaling in our patients might therefore disrupt the differentiation of naïve T cells into Th17 effector cells. The clinical features of patients 3 and 4 and other patients with AOID are similar to those with genetic defects of the Th1 response in which patients present with disseminated infection with mycobacteria of low virulence [19]. The presence of pustular reaction in a number of patients with AOID suggests a possible association between anti-IFN-γ autoantibody production and the recruitment of neutrophils (Figure 8). Recruitment of neutrophils into the psoriatic skin aggravates the symptoms because of the overproduction of pro-inflammatory cytokines, as observed by the significantly increased expression of IL1B, IL6, IL17, IL23, KRT17, and IFNG throughout the epidermis and dermis of patient 3 compared to normal skin tissue. It is noteworthy that depletion of neutrophils significantly relieves the symptoms [21]. The minor differences in protein staining found in the tissues from patients 1(GPP) and 3 (AOID) might have been due to the difference of the disease processes or the genetic backgrounds of the patients.

The pathogenetic mechanisms of anti- IFN-γ autoantibody production as a result of *TGFBR2* mutation is uncertain. However, marked increased expression of IFN-γ was observed in the lesions of patients 1 and 3. It is hypothesized that the *TGFBR2* variants in the patients caused dysregulation of T cell homeostasis, B-cell tolerance defects, and subsequent failure to counter-select developing autoreactive B cells, which promote the development of autoimmunity (AOID) through the presentation of self-antigen to T cells [22].

### 4.3. The Effects of Variants in Other Immunodeficiency Genes

Despite a reduction in TGFBR2 being consistent with a role in the pathogenesis of the pustular skin lesions in our four patients, it is possible that other factors contribute to the phenotypic presentation. With this consideration, we have noted the presence of rare variants in other immunodeficiency genes (*JAK2*, *JAK3*, *IL17RA*, *IL12RB2*, *CARD14,* and *TYK2*) in our various patients, which could feasibly contribute to the penetrance and/or differences in the clinical phenotypes of the patients (Table 1). Of note, the *TGFBR2* c.458del; p.Lys153SerfsTer35 variant was found in 17 individuals in our in-house exome database of 725 individuals of Thai ancestry with normal controls who had no history of pustular skin lesions. Thus, this variant may be a significant susceptibility factor and responsible for the higher prevalence of AOID in Thailand compared to the rest of the world [2]. Regardless, this is the first report to provide evidence for loss of function *TGFBR2* variant as a contributing or predisposing risk factor for GPP and AOID.

## 5. Conclusions

A frameshift mutation in TGFBR2 might be associated with GPP and AOID.A frameshift mutation in TGFBR2 is associated with overexpression of *KRT17* gene expression, a hallmark of psoriatic skin lesion.AOID might share pathogenetic mechanisms with GPP.

## Figures and Tables

**Figure 1 genes-14-00103-f001:**
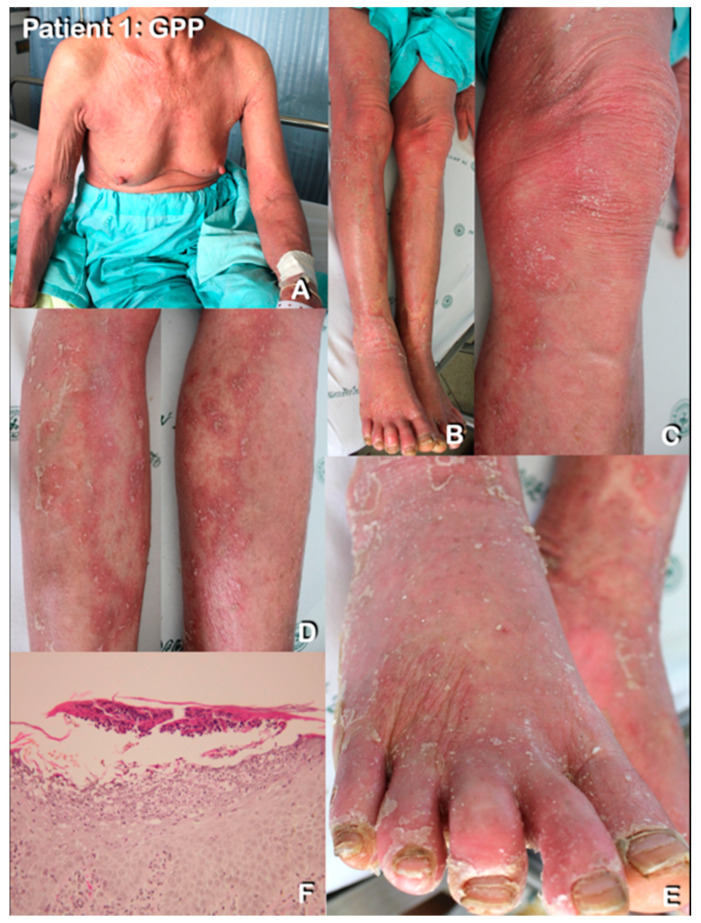
Patient 1 with GPP with pustular reaction. Clinical features and pathological findings of pustules in generalized pustular psoriasis. Extensive erythroderma on the chest, upper extremities (**A**) and lower extremities (**B**). Close-up view of both legs shows multiple non-follicular pustules which are notably observed at the edge of annular patches (**C**,**D**). Erythroderma on both feet with nail discoloration, subungual hyperkeratosis and onychodystrophy were noted (**F**). (**E**) Histopathological findings of a pustular lesion reveal subcorneal pustule and spongiform pustule in the upper layer of the epidermis.

**Figure 2 genes-14-00103-f002:**
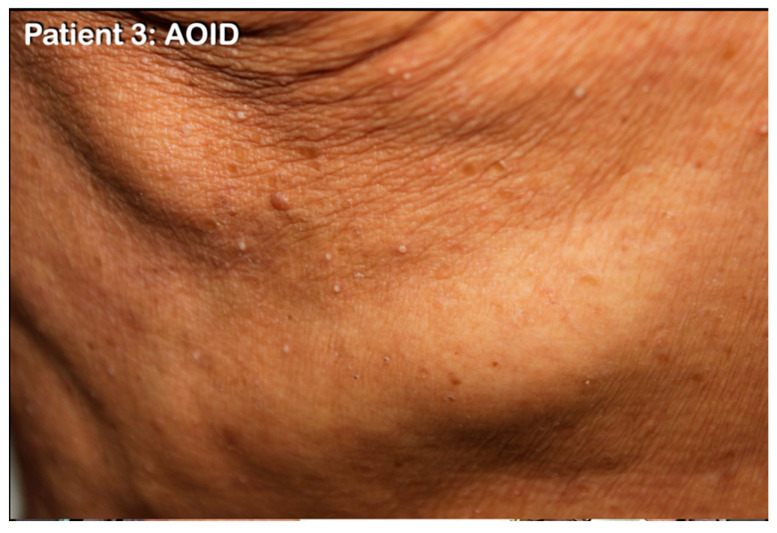
Patient 3 with AOID with pustular skin reaction. Non-follicular sterile pustules on the trunk.

**Figure 3 genes-14-00103-f003:**
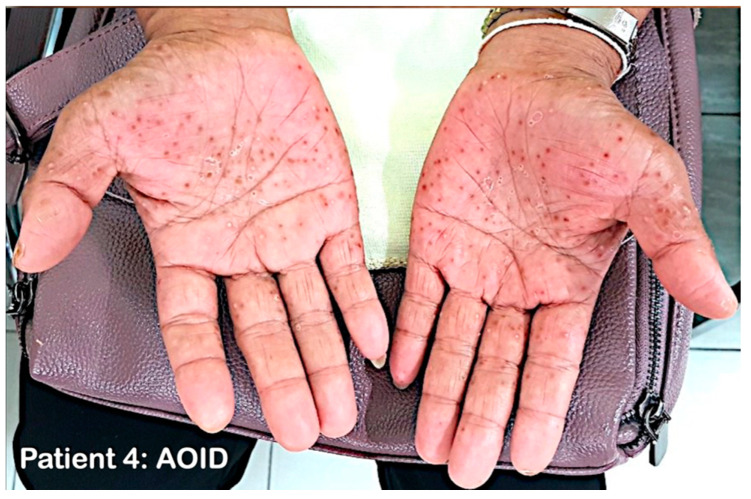
Patient 4 with AOID with pustular skin reaction. Discrete non-follicular sterile pustules on both palms one week after they first appeared.

**Figure 4 genes-14-00103-f004:**
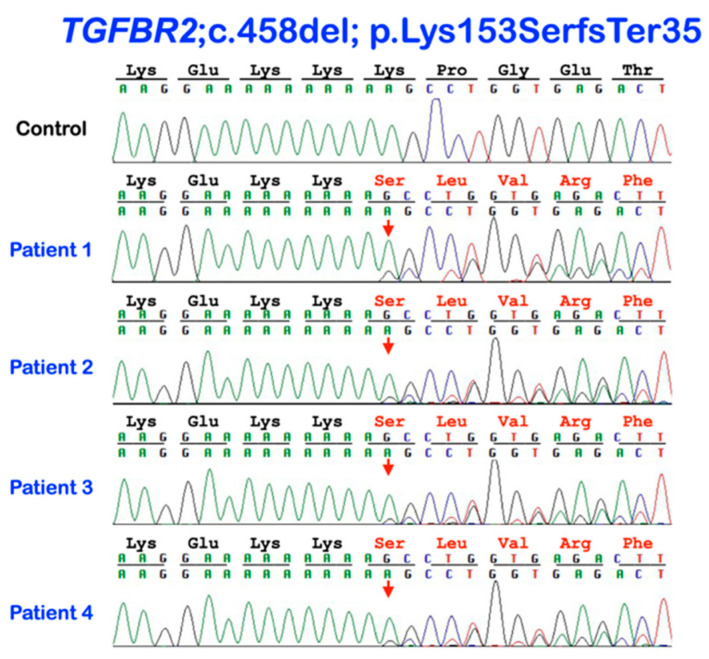
Electropherograms of the *TGFBR2* variant in patients 1–4. The heterozygous base deletion c.458del is predicted to cause premature truncation of TGFBR2 protein (p.Lys153SerfsTer35).

**Figure 5 genes-14-00103-f005:**
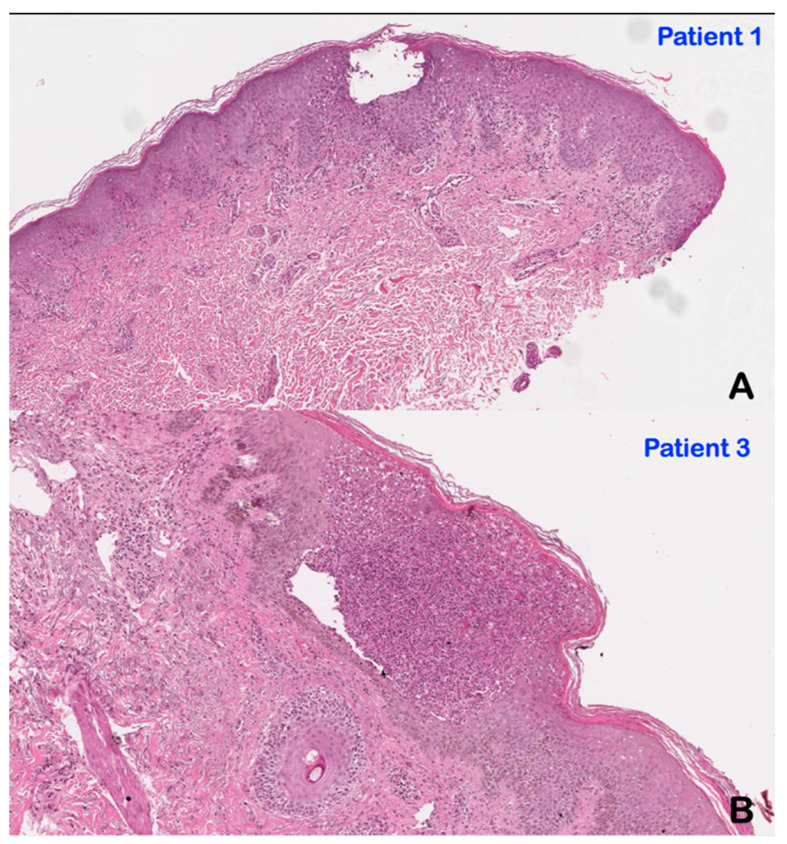
Histopathological findings of skin biopsies (**A**) Patient 1 (GPP). Note epidermal acanthosis, spongiosis, parakeratosis, and presented of subcorneal neutrophilic pustules (microabscess). Scattered neutrophils in the stratum spinosum are also seen. Upper portion of dermis shows perivascular lymphocytic and few neutrophils infiltration. (**B**) Patient 3 (AOID) Subcorneal and intraepidermal pustule containing large collections of neutrophils are observed. The contiguous epidermis shows acanthosis, spongiosis, and parakeratosis. Similar perivascular lymphocytes and neutrophils infiltration are observed in upper dermis.

**Figure 6 genes-14-00103-f006:**
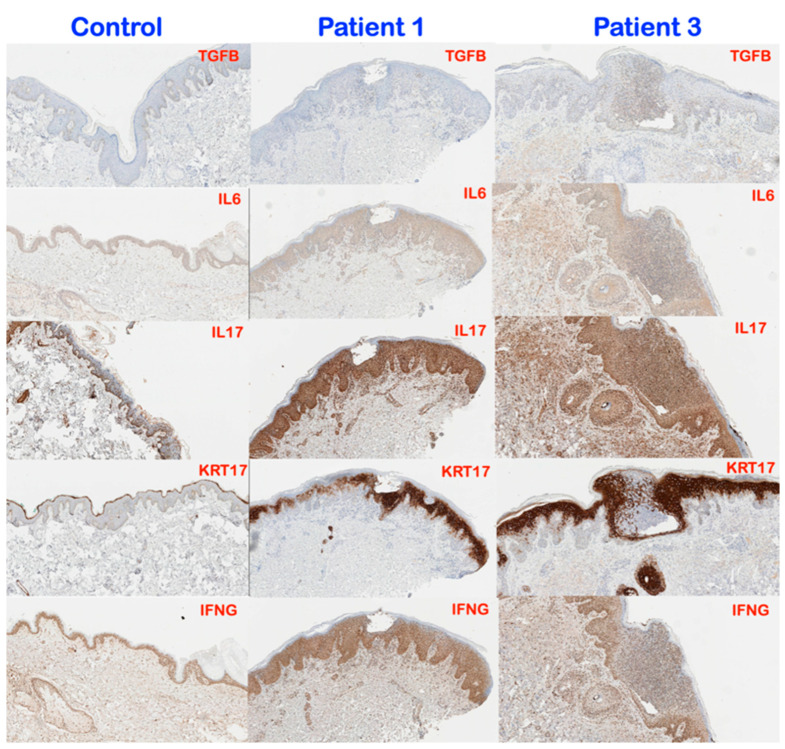
Representative images of immunohistochemistry for TGFB1, IL6, IL17, KRT17, and IFNG in the normal skin control, patient 1 (GPP), and patient 3 (AOID). TGFB1 expression is detectable at low levels in the epidermis and no immunoreactivity seen in dermis of normal skin tissue. No difference of TGFB1 expression between patient 1 and normal skin tissue, whereas TGFB expression is mildly detected within the collections of neutrophils in epidermis and some dermal cells in patient 3. Significant increased level of IL6, IL17, KRT17, and IFNG expression are seen throughout the epidermis and dermis of patients 1 and patient 3 compared to normal skin tissue. (Magnification, 8×).

**Figure 7 genes-14-00103-f007:**
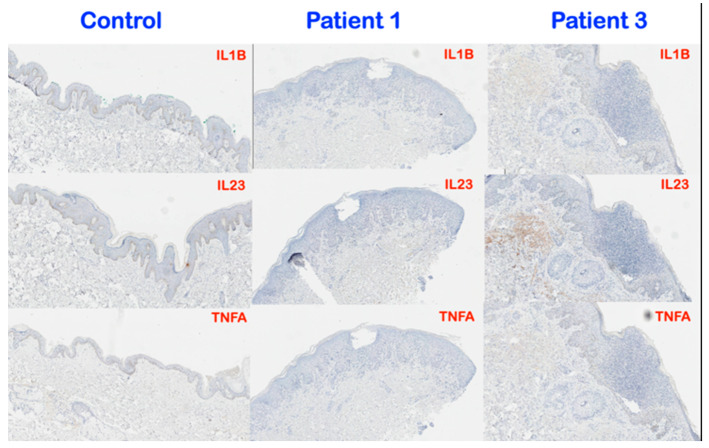
Representative images of immunohistochemistry for IL1B, IL23, and TNFA in the normal skin tissue, patient 1 (GPP), and patient 3 (AOID). IL1B and IL23 showed mildly increased expression in dermis of patient 3 whereas no difference between patient 1 and normal skin tissue. No significant difference of TNFA expression between normal skin tissue and patients 1 and 3 (Magnification, 8×).

**Figure 8 genes-14-00103-f008:**
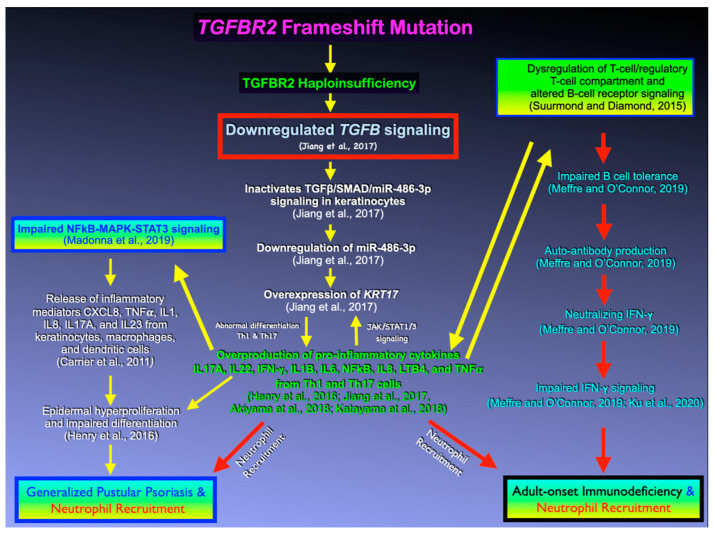
Hypothetical flowchart showing genetic pathways as a result of *TGFBR2* mutation. Downregulation of TGFβ signaling as a result of *TGFBR2* mutation leads to downregulation of the microRNA, miR-486-3p, overexpression of KRT17, over-production of pro-inflammatory cytokines, impaired NFkB-MAPK-STAT signaling, dysregulation of T-cells and B-cells and subsequent GPP and AOID with neutrophil recruitment [3,17,22,23,24,25,26,27,28].

**Table 1 genes-14-00103-t001:** Patients with *TGFBR2* variants and variants in other immunodeficiency genes.

Patients	Gender Age (year)	Diagnosis	*TGFBRII* Variant 1	*TGFBRII* Variant 2	*JAK2* Variant	*JAK3* Variant	*IL17RA* Variant	*IL12RB2* Variant	*CARD14* Variant	*TYK2* Variant
Patient 1	**Female** **(70 Yr )**	**GPP**	NM_001024847.2: **c.458del;** NP_001020018.1: **p.Lys153SerfsTer35;** rs79375991 3-30691880-AA-A (GRCh37)				NM_001289905.1: c.833G>A; NP_001276834.1: p.Arg278His; rs141467790; AF = 0.00128	NM_001258214.1: **c.52T>G;** NP_001245143.1: **p.Trp18Gly;** rs751550346; AF = 0.000003981	NM_024110.4; **c.2473G>A;** NP_077015.2; **p.Ala825Thr;** rs538251591	
Patient 2	**Female** **(24 year)**	**GPP**	NM_001024847.2: **c.458del;** NP_001020018.1: **p.Lys153SerfsTer35;** rs79375991 3-30691880-AA-A (GRCh37)	NM_001024847.2: **c.1019C>T**; NP_001020018.1: **p.Thr340Met;** rs34833812; AF = 0.001062 Het/Hom 4/0						NM_003331.5: **c.2977C>T**; NP_003322.3: **p.His993Tyr; rs201397594**; AF = 0.0001920
Patient 3	**Male** **(66 year)**	**AOID**	NM_001024847.2: **c.458del**; NP_001020018.1: **p.Lys153SerfsTer35;** rs79375991 3-30691880-AA-A (GRCh37)			NM_000215.3: **c.2678C>T**; NP_000206.2: **p.Pro893Leu**; rs772027199; AF = 0.000003978	NM_001289905.1: **c.679T>G**; NP_001276834.1: **p.Ser227Ala**; rs371494126; AF = 0.00001193			
Patient 4	**Female** **(41 year)**	**AOID**	NM_001024847.2: **c.458del;** NP_001020018.1: **p.Lys153SerfsTer35;** rs79375991 3-30691880-AA-A (GRCh37)		NM_001322194.1: **c.1174G>A**; NP_001309123.1: **p.Val392Met**; rs200018153; AF = 0.0006015					NM_003331.5:**c.2107C>T**; NP_003322.3:**p.Arg703Trp**; rs55882956; AF = 0.006692

## Data Availability

Not applicable.

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
