# Peer review of "Loss of Function TGFBR2 Variant as a Contributing Factor in Generalized Pustular Psoriasis and Adult-Onset Immunodeficiency"

_genes, 2022, doi:10.3390/genes14010103_

Round 1

Reviewer 1 Report

The authors describe a frameshift variant in the TGFBR2 gene as a possible causing factor underlying both generalized pustular psoriasis (GPP) and adult-onset immunodeficiency syndrome (AOID), also hypothesising that the two disorders may have a common pathogenic mechanism.

The article is clear and well-written and it provides a new insight on the genetic predisposing factors for GPP and AOID. I would however recommend that minor revisions are required. 

First of all, there is inconsistency in the presentation of clinical cases: while patient 4 history is throughly described (also with the detail of the precise hospital she was admitted to), patient 2 case description is very poor. I would suggest more uniformity in these descriptions. 

The conclusion paragraph also needs some slight amendendements:  as the authors state, the identified variant is also present in the healthy population and the number of patients described (4) is not sufficient to prove beyond doubt that this is a crucial predisposing factor on its own or that new pathogenic mechanisms linking AOID and GPP are certainly found. Therefore, I would change the first sentence of the conclusion paragraph from “A frameshift mutation in TGFBR2 is associated with GPP and AOID.” To “A frameshift mutation in TGFBR2 might be associated with GPP and AOID”, the same goes for the third point, that should be changed to “AOID might share pathogenetic mechanisms with GPP”.

Other considerations:

-       Figure 8 is quoted in the manuscript, but the picture itself is not present in the submitted file. 

-       According to standard nomenclature, gene names are to be written in italics to distinguish them from protein names, so all the times TGFBR2 is quoted as a gene, it needs to be italicized. 

-       Line 222: I would change “investigated by Sanger sequencing, suggestive of a polymorphism.” into investigated by Sanger sequencing, suggesting of a polymorphism”.

Author Response

Response to comments of Reviewer 1

Comments of Reviewer 1:

The authors describe a frameshift variant in the TGFBR2 gene as a possible causing factor underlying both generalized pustular psoriasis (GPP) and adult-onset immunodeficiency syndrome (AOID), also hypothesising that the two disorders may have a common pathogenic mechanism.

The article is clear and well-written and it provides a new insight on the genetic predisposing factors for GPP and AOID. I would however recommend that minor revisions are required. 

RESPONSE

Thank you for your kind words on our work.

First of all, there is inconsistency in the presentation of clinical cases: while patient 4 history is throughly described (also with the detail of the precise hospital she was admitted to), patient 2 case description is very poor. I would suggest more uniformity in these descriptions. 

RESPONSE

Thank you for this comment. We added more clinical detail and revised the description of patient 2 as follows:

A 27-year-old Thai woman had developed recurrent episodes of generalized pus-tular eruption for 3 years. She was then referred to Maharaj Nakorn Chiang Mai Hospital for definite diagnosis. She was afebrile and dermatological examination showed multiple non-follicular pustules on erythrodermic skin of her trunk and extremities. Microbiological studies from pustules failed to demonstrate any organisms. Diagnosis of relapsing GPP without systemic inflammation was made according to diagnostic criteria [Navarini et al., 2017]. The pustules were well-controlled with oral acitretin.

We also revised and shortened the clinical information of Patient 4 in this part as follows:

A 41-year-old, previously healthy Thai female, presented with a 6-month history of fever with generalized lymphadenopathy and weight loss. One week before admission at Srinagarind hospital, she developed pustular eruption on both hands (Fig. 3). Cervical lymph node biopsy revealed sclerotic lymphoid hyperplasia with plasmacytosis and a focus of eosinophilic abscess. Although no organisms were detected on histopathology, Mycobacterium abscessus was detected using PCR and isolated from cultures of the tissue. Gram and Ziehl Neelsen stains from pustules on the hands revealed no organisms and cultures were negative. The result of HIV test was negative, but a test for anti-IFN-γ autoantibodies was positive at 1:5000. The diagnosis of AOID with pustular eruption was made. Anti-mycobacterial therapy with azithromycin, levofloxacin, and prednisolone was initiated. Her symptoms, generalized lymphadenopathy, and pustular eruption improved after two weeks of treatment. Anti-mycobacterial therapy was continued for 20 months with remarkable clinical improvement.

The conclusion paragraph also needs some slight amendments:  as the authors state, the identified variant is also present in the healthy population and the number of patients described (4) is not sufficient to prove beyond doubt that this is a crucial predisposing factor on its own or that new pathogenic mechanisms linking AOID and GPP are certainly found. Therefore, I would change the first sentence of the conclusion paragraph from “A frameshift mutation in TGFBR2 is associated with GPP and AOID.” To “A frameshift mutation in TGFBR2 might be associated with GPP and AOID”, the same goes for the third point, that should be changed to “AOID might share pathogenetic mechanisms with GPP”.

RESPONSE

Thank you for these comments. We have corrected the conclusion as followed.

  1. Conclusion
  • A frameshift mutation in TGFBR2 might be associated with GPP and AOID.
  • A frameshift mutation in TGFBR2 is associated with overexpression of KRT17 gene expression, a hallmark of psoriatic skin lesion.
  • AOID might share pathogenetic mechanisms with GPP.

Other considerations:

Figure 8 is quoted in the manuscript, but the picture itself is not present in the submitted file. 

RESPONSE

Unfortunately the editorial staff who did the typeset for the manuscript left out Figure 8 and did not include it into the manuscript. That was why it was missing as you mentioned. It is included in this revised manuscript. We are sorry for the inconvenience.

Figure 8 is a hypothetical flowchart showing genetic pathways as a result of TGFBR2 mutation. This flowchart will help readers to picture that downregulation of TGFβ signaling as a result of TGFBR2 mutation leads to downregulation of the microRNA, miR-486-3p, overexpression of KRT17, over-production of pro-inflammatory cyto-kines, impaired NFkB-MAPK-STAT signaling, dysregulation of T-cells and B-cells and subsequent GPP and AOID with neutrophil recruitment.

According to standard nomenclature, gene names are to be written in italics to distinguish them from protein names, so all the times TGFBR2 is quoted as a gene, it needs to be italicized. 

RESPONSE

Thank you for this comment and we apologize for this oversight. We have corrected the gene and protein names according to your comment and HGMD (The Human Gene Mutation Database).

Line 222: I would change “investigated by Sanger sequencing, suggestive of a polymorphism.” into investigated by Sanger sequencing, suggesting of a polymorphism”.

RESPONSE

Thank you for this comment. It was corrected as you suggested:

Of note, the CARD14 variant c.2473G>A; p.Ala825Thr found in patient 1, who was from the Karen tribe, was also seen in 90 of 100 Karen tribe individuals we investigated by Sanger sequencing, suggesting of a polymorphism.

Reviewer 2 Report

Kantaputra et al. have revealed interesting findings related to the possible pathogenic mechanisms involved in generalized pustular psoriasis and adult-onset immunodeficiency. There are several things that could be improved, such as the following:

In the Introduction section, the authors should emphasize why they chose these two diseases to study mutations in TGFBR2 and clarify the aim of the study.

In subsection 2.1. Patients. The authors should explain more clearly why they present the four patients.

Line 99 - the authors should explain what ACH and PPP mean.

The Introduction section is not numbered.

Please explain abbreviations when first introduced in text (e.g. TGFBR2).

There are few data available on the Patient 2 in the manuscript, the authors should provide more data, such as how the diagnosis of GPP was made, the duration of the eruption. If there are available pictures of the clinical and/or histopathological appearance, they should be added.

In the case of  the Patient 3, the authors should specify the species of non-tuberculous mycobacterium that was isolated and how the diagnosis was made.

Line 139 - "anti-HIV test was negative" should be replaced with “the result of the HIV test was negative”.

Figure 8 is missing.

Author Response

Response to comments of Reviewer 2

Comments of Reviewer 2:

Kantaputra et al. have revealed interesting findings related to the possible pathogenic mechanisms involved in generalized pustular psoriasis and adult-onset immunodeficiency. There are several things that could be improved, such as the following:

In the Introduction section, the authors should emphasize why they chose these two diseases to study mutations in TGFBR2 and clarify the aim of the study.

RESPONSE

Thank you for this comment. We added the rationale of the study in the introduction as follows:

Perhaps reworded to?:

Previously, a genetic variant in SERPINA1 was found to be a predisposing factor in the presentation of both GPP and AOID [7]. Although TGFBR2 has not previously been reported to be associated with GPP and AOID, the expression of TGFBR2 is notably decreased or absent in psoriatic skin [8]. Therefore, we hypothesized that genetic variants in TGFBR2 might also predispose some patients in our cohort to GPP and AOID.

In subsection 2.1. Patients. The authors should explain more clearly why they present the four patients.

RESPONSE

Thank you for this comment. It was added in subsection 2.1 as you suggested.

Whole exome sequencing showed a heterozygous frameshift variant in TGFBR2 (c.458del; p.Lys153SerfsTer35) in two patients with GPP and two patients with AOID and they are present here.

Line 99 - the authors should explain what ACH and PPP mean.

RESPONSE

Thank you for this comment. That sentence has been corrected as followed. Inclusion criteria were patients with AOID or pustular psoriasis phenotypes, including GPP, acrodermatitis continua of Hallapeau (ACH), and palmoplantar pustulosis (PPP).

The Introduction section is not numbered.

RESPONSE

Thank you for this comment. It was corrected as 1. Introduction.

Please explain abbreviations when first introduced in text (e.g. TGFBR2).

RESPONSE

Thank you for this comment. We added the abbreviation explanations for ACH, PPP, TGFB, TGFBR2, and TGFBR1.

There are few data available on the Patient 2 in the manuscript, the authors should provide more data, such as how the diagnosis of GPP was made, the duration of the eruption. If there are available pictures of the clinical and/or histopathological appearance, they should be added.

RESPONSE

Thank you for this comment. This request was also made by Reviewer 1 and was addressed as described earlier.

Unfortunately, we tried our best in archive review, but the pictures of the clinical or histopathological study are not avaiable.

In the case of the Patient 3, the authors should specify the species of non-tuberculous mycobacterium that was isolated and how the diagnosis was made.

RESPONSE

Thank you for this comment. The diagnosis of possible disseminated non-tuberculosis mycobacterial (NTM) infection in Patient 3 was evidenced by histological findings from lymph node biopsy showing granulomatous lymphadenitis and responsiveness to NTM treatment regimen. The microbiological studies failed to demonstrate any organisms.

We added more clinical detail and revised the description of Patient 3 as follows:

A 66-year-old man presented with generalized lymphadenitis as a result of possible disseminated non-tuberculosis mycobacterial (NTM) infection (evidenced from lymph node biopsy revealing granulomatous lymphadenitis and responded well to NTM treatment regimen).

Line 139 - "anti-HIV test was negative" should be replaced with “the result of the HIV test was negative”.

RESPONSE

Thank you for these comments. It was corrected as followed.

The result of HIV test was negative.

Figure 8 is missing.

RESPONSE

Thank you for this comment. Unfortunately, the editor staff who did the typeset for the manuscript left out Figure 8 and did not include it into the manuscript. That was why it was missing as you mentioned. It is included in this revised manuscript. We are sorry for the inconvenience.

Reviewer 3 Report

This is a well-written manuscript about the loss of function TGFBR2 variant as a contributing factor in generalized pustular psoriasis and adult-onset immunodeficiency.

Although four patients are too few to come to any definitive conclusions on the role of TGFBR2 in these diseases, I believe it is a good contribution and an interesting message to send to the scientific community.

The paper is well constructed, and the experimental study design was well conducted.

Minor revisions:

-        The name of the genes should be written in italics

-        Page 2, line 69: it would also be interesting to mention this work: Genovese G, et al. Pustular Psoriasis: From Pathophysiology to Treatment. Biomedicines. 2021 Nov 23;9(12):1746. Doi: 10.3390/biomedicines9121746.

-        I could not see figure 8. In any case, the figure could be briefly explained in a separate chapter.

-        It would be interesting and useful for the readers to add a conclusive table summarizing the genetic defects found in both GPP and AOIDS, including new findings.

Author Response

Response to comments of Reviewer 3

This is a well-written manuscript about the loss of function TGFBR2 variant as a contributing factor in generalized pustular psoriasis and adult-onset immunodeficiency. Although four patients are too few to come to any definitive conclusions on the role of TGFBR2 in these diseases, I believe it is a good contribution and an interesting message to send to the scientific community. The paper is well constructed, and the experimental study design was well conducted.

RESPONSE

We would like to thank Reviewer 3 for the kind words on our work. You made us very very happy.

Minor revisions:

The name of the genes should be written in italics

RESPONSE

Thank you for this comment and we apologize for this oversight. We have corrected the gene and protein names according to your comment and HGMD (The Human Gene Mutation Database).

Page 2, line 69: it would also be interesting to mention this work: Genovese G, et al. Pustular Psoriasis: From Pathophysiology to Treatment. Biomedicines. 2021 Nov 23;9(12):1746. Doi: 10.3390/biomedicines9121746.

RESPONSE

Thank you for this comment. This is a great paper on pustular psoriasis. Thank you for your advice. It is added as a reference.

I could not see figure 8. In any case, the figure could be briefly explained in a separate chapter.

RESPONSE

Unfortunately, the editorial staff who did the typeset for the manuscript left out Figure 8 and did not include it into the manuscript. That was why it was missing as you mentioned. It is included in this revised manuscript. We are sorry for the inconvenience.

Figure 8 is a hypothetical flowchart showing genetic pathways as a result of TGFBR2 mutation. This flowchart will help readers to picture that downregulation of TGFβ signaling as a result of TGFBR2 mutation leads to downregulation of the microRNA, miR-486-3p, overexpression of KRT17, over-production of pro-inflammatory cytokines, impaired NFkB-MAPK-STAT signaling, dysregulation of T-cells and B-cells and subsequent GPP and AOID with neutrophil recruitment.

It would be interesting and useful for the readers to add a conclusive table summarizing the genetic defects found in both GPP and AOIDS, including new findings.

RESPONSE

Thank you so much for this comment. However, we feel that the genetic defects for GPP and AOID are covered in many papers such as Genovese et al., 2021. Our intention is to report of the new gene (TGFBR2), not new clinical findings. Therefore, we did not include a table on that. We hope you understand our point. We feel that what we illustrate in the flowchart in Figure 8 is sufficient for the readers to understand the pathogenesis of GPP and AOID.

Round 2

Reviewer 2 Report

The manuscript has been sufficiently improved and most of the comments were addressed in the new version of the article.